# Characterization of Commercial Gas Diffusion Layers (GDL) by Liquid Extrusion Porometry (LEP) and Gas Liquid Displacement Porometry (GLDP)

**DOI:** 10.3390/membranes12020212

**Published:** 2022-02-11

**Authors:** René I. Peinador, Oumaima Abba, José I. Calvo

**Affiliations:** 1Institut de la Filtration et des Techniques Séparatives (IFTS), Rue Marcel Pagnol, 47510 Foulayronnes, France; 2Département de Génie des Procédés et Bioprocédés, Faculté de Science et D’ingénierie (FSI), Université Toulouse III-Paul Sabatier, 31400 Toulouse, France; abba.oumaima97@gmail.com; 3Departamento de Física Aplicada, Escuela Técnica Superior de Ingenierías Agrarias (ETSIIAA), Universidad de Valladolid, 34071 Palencia, Spain; jicalvo@termo.uva.es; 4Surfaces and Porous Materials (SMAP), Associated Research Unit to CSIC, UVainnova Bldg, Paseo de Belén 11, 47071 Valladolid, Spain; 5Institute of Sustainable Processes (ISP), Dr. Mergelina s/n, University of Valladolid, 47071 Valladolid, Spain

**Keywords:** membrane characterization, pore-size distribution (PSD), capillary pressure, GDL, LEP

## Abstract

This works aims to study the porous characterization of several commercial Gas Diffusion Layers (GDL). Three carbon-based porous GDL featuring a highly rigid microstructure of interconnected pores of several manufacturers were analyzed. Gas Liquid Displacement Porometry (GLDP) and Liquid Extrusion Porometry (LEP) have been used to obtain their pore size distributions (PSD) and the mean and mode pore diameters (*d_avg_* and *d_mod_*), by means of a gas/liquid and extrusion porometer developed at IFTS. N-dodecane liquid has been used to completely wet the GDL’s assuring penetration of the liquid into the carbon fibrous structure. The results demonstrated the accuracy of the setup on characterizing GDL in the Particle Filtration (PF) range by GLDP and LEP, with reasonable agreements of resulting PSD and average sizes between both techniques when GLDP and LEP results are compared. Differences can be explained in terms of the high pore connectivity of these kinds of structures.

## 1. Introduction

Climate change is a global problem that is widely recognized by the international scientific community. One of the main causes that contribute to this change is the emission of greenhouse gases, particularly CO_2_ as a consequence of human economic activity. Faced with this climate emergency, the Paris Agreement proposes a drastic reduction in the sources of CO_2_ creation, especially fossil fuels, which are responsible for much of this degradation. In particular, it is intended to replace fossil fuels in fields as important as transport or heating with ecologically acceptable alternatives. This seems to be a difficult task considering the always increasing demand of energy consumption worldwide. It is estimated that global energy demand at 2040 will rise by a 37% than compared with 2014 levels [1], making it increasingly complicated to match such increasing demands with sustainable and environmentally friendly practices. In this sense, fuel cells are called to play a very important role in the coming years.

A fuel cell (FC) is a device capable of directly converting chemical energy into electrical energy by means of a highly efficient electrochemical process with minimal environmental impact. Currently, there is a wide variety of fuel cells in different stages of development, and they have wide potential applications in many fields such as transport, fixed, and portable power supplies [2,3]; with applications to vehicles [4,5,6]; satellites [7]; telecommunication stations [8,9,10]; and remote area power supply [11,12,13,14,15] to name a few [16] based on factors such as high energy-conversion efficiency and environmental cleanness, along with quick start up and simple design [17,18,19,20,21,22,23,24].

On the other side, FC can be classified according to many aspects, including the following: type of fuel and oxidant combination, type of electrolyte used, operating temperature, its efficiency or power, and type of use and catalyst used. Nevertheless, the most common form of classification is by the type of electrolyte used. Among them, there are two that use polymeric selective membranes as electrolytes, which are as follows: The Proton Exchange Membrane Fuel Cell (PEMFC) and the Direct Methanol Fuel Cell (DMFC).

Basically, the membrane forms part of the Membrane Electrode Assembly (MEA), a key part of the system which comprises the following: two gas diffusion layers (GDL), two catalyst layers, and a proton exchange membrane placed in between [25]. GDLs provide mechanical support for the catalyst and membrane layers while serving to control mass, heat, and electron transport [24].

In order to achieve these tasks, GDL must fulfill some specific properties in their design: gas and vapor diffusivity and permeability, porosity, surface contact angle, electrical and thermal conductivity, mechanical strength, and durability at the expected operating conditions [16,26,27,28]. Proper design of GDL allows them to fulfill their main role: provide high electrical and thermal conductivity and chemical/corrosion resistance, in addition to controlling the proper flow of reactant gases (hydrogen and air) and managing water transport out of the membrane electrode assembly (MEA). Additionally, this layer must also have controlled compressibility to support external forces from the assembly and not deform into bipolar plate channels to restrict flow [29].

Mostly GDL are designed as a combination of two distinct layers: a backing macroporous substrate (MPS), which corresponds to the actual GDL, and a microporous layer (MPL), sometimes used to improve GDL characteristics. The MPSs are usually made from carbon black (powder) with the addition of some polymer used as binder and hydrophobic agent [24].

Graphitized carbon-fiber is an ideal material for MPS of GDLs because it is strong, light, has high electrical conductivity, and because it is chemically inert and stable inside fuel cell. GDLs are typically between 150 and 400 µm thick with porosity between 70 and 90%. The fibers are about 10 microns in diameter. An ideal GDL should possess hydrophobicity so that water does not wick and spread throughout the pore structure; high effective gas diffusivity which requires high porosity and low tortuosity; and high permeability to support liquid water capillary flow.

In order to minimize the contact resistance between the GDL and catalyst layer, limit the loss of catalyst to the GDL interior, and help to improve water management, some GDLs are supported by an MPL (Microporous Layer), which is usually fabricated by intermixing a hydrophobic agent, polytetrafluoroethylene (PTFE), with carbon black, which is located between the catalyst layer and the proper GDL.

Park et al. [30] reviewed single- and dual-layer GDLs with various types of carbon materials and approaches with a variety of carbon- and metal-based macroporous substrates using different fluorinated hydrophobic polymers [31].

The performance of membrane-based FC is closely related with water and gas transport of GDL layers, which should require a balanced hydrophobicity and hydrophilicity properties. This makes carbon paper GDL substrates very interesting as that their porosity can be controlled. For example, higher porosity GDLs could result in membrane degradation due to a decrease in electrical conductivity and, consequently, induce performance degradation [32]. Normal values of GDL porosity range 31.8% to 73.9% [33].

Studying the porous properties of the GDL is a challenge because most of the existing techniques used for conventional porous structures are not suitable since GDLs display many atypical characteristics (they are thin, highly porous, compressible, electrically conductive, neutrally wettable, anisotropic, and so on).

As such, experimental techniques must generally be developed that are adapted to atypical features unique to the GDL.

Among the techniques used to obtain information about the porosity and pore size distribution of GDL are intrusion methods such as Mercury intrusion and Capillary flow porometry [31].

They all make a general assumption that the capillary structure can be represented by a bundle of tubes, which is hardly considered with respect to parallel pores with a particular range of radii. This simplifying assumption does not accurately capture the complex pore structure of a GDL, which is a system of more or less interconnected pores. Inferences about the internal structure drawn from such measurements should be made with careful consideration. New methods, such as hydraulic admittance [34], are under development, which may result in more accurate information about complex pore structures found in GDLs [31].

Several studies [35] have tried to model the complex porous structure of GDL membranescarbon fibers. Many of these simulations, based on the classical invasion-percolation algorithm [36], combine the tracking of fluid–fluid interfaces and the invasion of throats. A recently published paper by one of the authors has analyzed, from a pore network simulation (PNS), the accuracy of PSD obtained for porous films based on the assumption of a bundle of cylindrical parallel straight through pores. Authors depart from two or three-dimensional networks, for which their characteristics are known a priori and neglect liquid trapping [37], to obtain an estimation of throat size distribution (TSD) by comparing these results with experimental input showing, in some cases, a significant discrepancy with input data. From this study, the procedure based on the capillary pressure curve combined with the Young–Laplace equation (basis of LEP technique) results in a better determination of the mean radius [38].

The ratio of hydrophobic and hydrophilic pores could be characterized by using a combination of the results from Capillary Flow Porometry (CFP), Hg intrusion Porosimetry (HgP), and water intrusion porosimetry [39].

Gas-Liquid Displacement Porosimetry (GLDP), also termed CFP, is a characterization method that can provide the complete Pore Size Distribution (PSD) of filtration membranes having pores in the micrometric range. It has been used very often to determine pores from 50 nm to tens of microns, and its reliability and accuracy have provided the technique with the status of recommended standard for MF membranes. On the contrary, Liquid Extrusion Porometry (LEP), which also allows the determination of the PSD of membranes in this range of pores, has less acceptance among researchers in part due to the lack of commercial equipment that allows this type of analysis automatically and precisely. However, LEP can provide important complementary information on the wettability of pores present in the sample, information that is crucial for the application of GDL designed for membrane-based FC.

The aim of the present study is to compare the characteristics and outputs of two techniques frequently used to characterize porous structure of membranes and porous layers. In this case, GLDP and LEP will be used to determine the PSD of several carbon fibers based GDL’s. The results will be compared and information about wettability and pore interconnection of the analyzed GDL will be extracted from such comparison.

## 2. Materials and Methods

### 2.1. GDLs

Three flat-sheet GDLs made from carbon paper rigid fibbers with no polymeric binder addition and supplied from recognized manufacturers have been characterized. Names of suppliers and models are provided in Table 1.

All GDLs were obtained from each manufacturer in the form of flat carbon sheet and were tested in a specific 25 mm diameter housing cell. All samples were dried out and immersed into the porosimetric wetting liquids for half an hour under vacuum pressure (200 mmHg) at room temperature to assure complete fibers soaking. Each GDL was tested using three different samples taken from the same batch. The results were averaged, and the standard deviation of results was calculated; finally, the percentage of error was calculated as Standard deviation/Average value.

All these carbon-based GDLs present a quite similar structure with a set of cross-linked carbon fibers, which leave interstices in between that give rise to the various pores present on the layer. This can be clearly observed in Figure 1, where an SEM image of one of the GDLs studied, AVCARB, is presented. The image shows perfectly how carbon fibers that make up the give way to a series of strongly interconnected pores.

### 2.2. Gas Liquid Displacement Porometry (GLDP) Method

The basis and fundamentals of GLDP have been previously explained in detail [40]. GLDP measurements were performed with IFTS Capillary Flow Porometer (CFP) (model IFTS-PRM-8720^®^, Foulayronnes, France) consisting of an automated pressure constant device suitable for working in Gas/Liquid and Extrusion Porometry configurations. The device is designed for testing pore sizes down to 0.3 µm and up to 500 µm. It uses relatively very low pressures from 0.5 kPa (minimum) up to a maximum of 200 kPa for the characterization of porous or fibers GDL in the low MF to Particle Filtration range. The equipment allows implementing very stable pressure (accuracy ± 0.1 mbar), which results in very accurate measurement of resulting fluxes by mass flowmeter (accuracy ± 1 mL/min). The included software is able to determine several important parameters related with Pore size characterization, including mean pore diameter, peak pore size, PSD, fluid permeability and bubble point. It also can be adapted to analyze various samples for membrane filtration configurations or modules as hollow fiber, tubular, and flat sheet.

The Gas-Liquid Porometer is able to obtain the PSD of the GDL by using the capillary principle. The filter media being previously wet in a liquid (wetting phase) fills all pores present in the sample. Then, upstream pressure of displacing gas is increased at a predetermined rate, and gas flow downstream the media is observed, indicating the passage of displacing gas through all pores yet opening in GDL, starting at the GDL maximum pore diameter (the well-known concept of Bubble point for Gas-Liquid) until flow is stablished thought all the pores, including the smallest ones.

From an experimental point of view, the GLDP method used for the characterization of GDL is very similar to LEP. The sample is first saturated in a high wetting liquid, which has low surface tension, low vapor pressure, low viscosity (~1 cp), and presents chemically low interaction with carbon based material. In this study the wetting liquid was n-Dodecane (99% purity, anhydrous) supplied by Thermofisher^®^, (Paris, France) and possessed a surface tension value of 25, 35 mN/m. For GLDP, the displacing fluid consists in a gas steeply pressurized starting for a very low pressure; then, the gas flow across the GDL is monitored. As pressure progressively increased beyond the bubble point, successive pores of decreasing sizes gradually empty and contribute to gas flow *J_k_* through GDL until all pores become empty from wetting fluid and successive flows becomes proportional to the pressure. The differential air permeability contribution at each pressure step can be plotted in terms of the size of pores, yet opened by using the same Equation (1), and then obtaining the corresponding PSD, similarly to Equation (2).

### 2.3. Liquid Extrusion Porometry (LEP) Method

LEP is a variation of GLDP reviewed by Jena and Gupta for PMI [41]. The idea is to minimize difficulties on measuring the gas flow through the filter by introducing a new porous membrane as a substrate of the one to be analyzed. This new substrate must have pores smaller than all those present in the original filter. In such a manner, the air used to expel wetting liquid from the filter analyzed cannot pass through and the flow of air is substituted by measurement of the mass of liquid drained from emptied pores. Similarly, fundamentals of LEP can be followed in Peinador et al. [40]. In order to obtain capillary pressure curves, the same commercial device, IFTS Fluid Porometer, used in previous GLDP experiences was used, but it is now docked to an external module consisting of an appropriate sample holder, where the sample to be analyzed is placed onto a filtration membrane material with much smaller pores and a precise analytical balance (Figure 2). The role of the CFP equipment was now to supply a stable pressure above the sample, while the balance was able to monitor the drained mass from the wetted media. This IFTS designed commercial setup allows converting in a dual GLDP/LEP equipment to accomplish such LEP experiments; therefore, it can be used to characterize Gas Diffusion Layers (GDL) integrated into Polymer Electrolyte Membrane Fuel Cells (PEMFC) [22,23,24,25,26,27,28].

The experimental procedure is quite similar to that used for GLDP analysis. For the generation of capillary pressure curves, it was the drainage and further imbibition of wetting liquid. Using the CFP Porometer, the capillary pressure is slowly increased, and the wetting liquid flows out of the sample onto an analytical balance (accuracy ±0.1 mg). As the system reaches equilibrium condition *(*Δ*m/*Δ*t*~0*),* at each capillary pressure, the volume (mass) of liquid drained is recorded before the pressure is increased again. Once the maximum desired pressure has been reached, capillary pressure is slowly relieved and the wetting fluid flows back into the sample. Using these data, a capillary pressure vs. drained mass curve can be generated.

The capillary pressure can be defined from the next equation [26].
*Pc* = *P* (*wetting liquid*) − *P* (*non*-*wetting fluid*) = *ρgh* + *P_Atm_* − *P_G_*
(1)

Gas pressure is *P_G_* (Pa)*,* controlled by the porometer above the sample; and *h* (cm) is the distance between the liquid reservoir level and the sample surface, which is roughly 5 cm. Therefore, during the experiment, there is a small column of liquid open to barometric pressure *P_Atm_* (Pa) and acting on the surface of the liquid reservoir. To correctly apply the LEP technique, it is necessary to place a capillary barrier under the sample, which must be a non-porous material or, at least, possess clearly narrower pores than the sample to be analyzed. In our case, a hydrophilic CN (cellulose nitrate) porous membrane supplied by Sartorius^®^ (Gottingen, Germany) was used in the form of flat discs with a diameter Ø = 25 mm (nominal pore size: 0.45 µm).

The main characteristics required of the wetting liquid used in LEP experiments are very similar as those used for GLDP. In addition, ideal flow properties (low viscosity) and low propensity for evaporation at room temperature are requited. Evaporation is not desirable as it can result in pores in both the sample and the hydrophilic barrier membrane becoming not wet enough, allowing for air to breakthrough. Accordingly, in all LEP experiences, the wetting liquid was the same as the one used in GLDP experiences (n-Dodecane, surface tension 25.35 mN/m).

The plot of contributions to total mass drained was considered only for positive capillary pressures of the drained liquid for each opened pore and converted into PSD, in terms of normalized differential mass drained divided by differential diameter *dm*/*dD* (mg/µm) (%).
(2)$${\left(\Delta m/\Delta d\right)}_{k}\left(\%\right)=\frac{{\left(\Delta m/\Delta d\right)}_{k}\;\;}{\sum^{}{\left(\Delta m/\Delta d\right)}_{k}}\times 100$$

The pore sizes obtained by LEP were average *d_p_* (m) from the gas capillary pressure curve.

## 3. Results and Discussion

### 3.1. Flux vs. Pressure for GLDP Technique

The PSD of three GLD were obtained via GLDP, depending on the size of pores population for each sample. Gas Liquid Porometry, as previously commented, is based on the effluent (i.e., flux-pressure) curve obtained when consecutive pores of the GDL are successively subjected to flow of the displacing gas (air). The resulting curve is expected to be S-shaped, with the maximum slope corresponding to the moment all pores are opened to flow so that the permeability became constant and slightly changing with pressure (for GLDP, corresponding to a compressible displacing fluid (Figure 3a).

### 3.2. Drained Mass vs. Capillary Pressure for LEP Technique

Similarly, PSD was characterized via LEP based on the extrusion mass of the wetting liquid by air (non-wetting fluid) for the case of GDL. As previous commented in the LEP technique section, the porometer compresses the gas above the sample, thereby changing capillary pressure. After each change in gas pressure and, therefore, capillary pressure, the mass of the liquid on the balance is monitored and the corresponding changes are recorded. The system is held at a constant gas pressure until the wetting fluid (n-dodecane) mass reading on the balance becomes stable. The resulting curve is an increasing one (similar to those presented in Figure 3b), until all contained masses of the liquid are drained or extruded from the media and an almost constant plateau is achieved.

### 3.3. Comparison of Pore-Size Distributions (PSDs)

For each GDL and technique, three repeated runs were carried out and results were averaged.

Figure 3 displays PSDs and porometry run curves for the AVCARB GDL obtained via GLDP and LEP techniques, respectively. The pore diameters measured were in the range under 100 µm, and the average pore diameter *d_avg_* was 19.4 µm with good reproducibility (error ~1%) for GLDP technique. On the other hand, for the case of LEP tests, the pore diameters measured were in a similar micrometric range, with an average pore diameter and error percentage: *d_avg_* = 26.3 ± 3%, respectively. Remember that *d_avg_* can be considered a reliable estimation of mean pore size then allowing to estimate the size of particles typically retained in cases of membrane filtration. Then, key observations for this sample are the following: (*i*) repeated PSDs were similar for each technique, indicating good reproducibility and, thereby, reliability between both techniques. It is worth noting that, even if LEP runs were quite close each to other, the translation into PSD resulted in some discrepancies, which still are perfectly assumable. These run-to-run discrepancies in GLDP runs are apparently higher, but since they mostly affect the final permeability and not the pressure corresponding to the change or curvature in the experimental curve, these values result in similar peaks and then to a closer *d_avg_* values.

In any case, both techniques present a remarkable reproducibility and fair low errors, particularly taking into account the intrinsic variability of filter media.

The next figure shows the experimental runs and the corresponding PSDs for the SPECTRACARB GDL as obtained from GLDP (Figure 4a) and LEP (Figure 4b).

The *d_avg_* value was 26.6 ± 6.9% µm for GLDP, while, for the case of the LEP technique, PSDs resulted in an average for pore diameter of 47.3 µm, which is remarkably higher than that obtained from GLDP and also had higher discrepancies between runs (~13% for LEP and 7% for GLDP).

Finally, the results coming from the GLDP technique for SIGRACET GDL samples are shown in Figure 5a. In this case, the averaged value was *d_avg_* = 30.4 ± 29% µm, showing clearly less reproducible runs. The results obtained for LEP (Figure 5b) presented, for this fiber, less variability, with a mean value of *d_avg_* = 42.1 ± 0.9% µm. This sample-to-sample variability, which is higher for SIGRACET than for any other GDL studied (as can be found in Table 2), could be attributed to different fabrication methods and the not-so strict control of fabrication parameters.

The results of *d_avg_* for all GLD studied and both techniques are summarized in Table 2, along with corresponding error ranges. What is more puzzling, apart from certain sample variability that has yet been commented upon, is the overestimation of *d_avg_*, which results from LEP runs as compared with GLDP ones.

This fact is reflected in *d_avg_* values clearly higher for LEP than for GLDP (from 35% higher for AVCARB to 77% in the case of SPECTRACARB). Sometimes, these differences could be attributed to a not properly finished experience, which could result in a more or less extended log-normal PSD and, consequently, changes in the averaged pore size (similar peaks could result in different averaged values for highly skewed distributions). To check such a hypothesis, a more frequent value of each PSD (*d_mod_*), i.e., the mode, has been obtained, and the corresponding results are averaged for GDLP sand LEP. The resulting values are also found in Table 2.

Currently, the values of mode pore size decreases for all experiences (as expected). Nevertheless, the mean mode values for LEP are still clearly higher than those for GLDP (except for SIGRACET, where differences are much less noticeable, around 2%). Thus, it must be concluded that LEP tends to overestimate mean pore size values than compared with GLDP results.

The reason for this difference should not lie in the lower precision of the LEP technique. This can be inferred by taking into account that LEP experiences are quite precise and reproducible, with reproducibility comparable to that obtained with GLDP. On the other hand, the theoretical bases of LEP analysis are practically the same as those of GLDP, and the experimental precision of the appropriate technique can obtain reliable results. The reason for this discrepancy may be in the interconnectivity of the existing pores in this type of filter. Said interconnectivity (which can be clearly observed in Figure 1 for the AVCARB filter) is common to all GDLs due to their own manufacturing method. Simulation studies using PNS modelling predict that, for filters presenting highly interconnected pores (such as those studied here or face masks, in which similar results were observed [40]), the LEP technique should produce higher values than a technique such as GLDP based only on flow data and, consequently, is more related to the narrowest section found in a given pore.

## 4. Conclusions

GLDP and LEP techniques, respectively, provide the PSD of the analyzed filters (flow-based data for GLDP and drained mass-based for LEP), both after applying the same Young–Laplace equation to convert experimental pressure data into pore diameters. Obviously, the experimental behavior of each technique and, consequently, the resulting PSD could be slightly different depending on the interaction of the displacement/extrusion liquid with each GDL sample but also depending on the actual microstructure or pore/fibers connectivity of the GDL, which varies from one to another type. The aim of this study is then to compare such results directly in the possible range (principally in particle filtration).

Three series of carbon based commercial GDL have been analyzed using a precise, accurate, and fast automated GLDP/LEP commercial device. The device designed and marketed by IFTS is an accurate CFP (GLDP) porometer, which has been improved principally in low pressure measurements by connecting it to an analytical balance and proper LEP cell. The setup is able to analyze filters ranging from 0.5 µm to Particle Filtration range. Both techniques (GLDP and LEP) were developed to provide information about the PSD of GDL media and make use of the same Young–Laplace equation to convert experimental data into pore size values. The techniques differ in the role of wetting/pushing gas in the process of emptying pores from a previous filling fluid.

The results showed a nice agreement between different runs for both techniques and also reasonable agreement was found when comparing GLDP and LEP outputs, except for SPECTRACARB GDL media. Certainly, broader PSDs are found from LEP technique (which, in turn, is able to discriminate larger pores contributing to mass) than for GLDP runs. As a result, the information on PSD and, mainly, the mean pore sizes deduced from it must be assessed and questioned.

In that sense, it should be remarked that (for the three filters analyzed and both mean or mode values) the resulting curves from LEP are slightly shifted to higher pore sizes than those coming from GLDP. This fact can be related with high pore interconnectivity that can be expected in this kind of filters (see Figure 1). This result has been predicted by PNS modeling. In the case of PNS applied to a bunch of parallel tubes, the simulation explains that data based on the capillary pressure curve (i.e., Liquid Extrusion Porometry) combined with the Young–Laplace equation results in better determinations of the mean radius than the procedure based on normalized flow rate curves. However, all invaded pores at a given pressure step in both procedures are assigned to one pore size corresponding to the specified capillary pressure regardless of their actual sizes. Thus, these simple procedures do not make any distinction between TSD and PSD [38,41].

In conclusion, we can conclude that an accurate LEP technique measuring air–liquid capillary pressure curves provides essential information about a porous/fibrous material such as porosity, breakthrough pressure (which provides information about maximum pore sizes present in the sample, similarly to bubble point), fluid–solid wettability, and, obviously, PSD. The comparison of such PSDs obtained for GDLs with results coming from GLDP technique allows obtaining insights about the inner connectivity of porous structure of the pores/throats and saturation relationship, which depends on the interaction of fluid–solid wettability, and these constraints very often scarcely resemble the capillary pore model assumed by GLDP.

In this sense, the combination of both techniques is critical to understand the fuel cell performance based on how GDL porosity/PSD of the fiber structure has a direct effect on the MEA assembly, by characterizing the contact of GDL with the polymeric membrane.

## Figures and Tables

**Figure 1 membranes-12-00212-f001:**
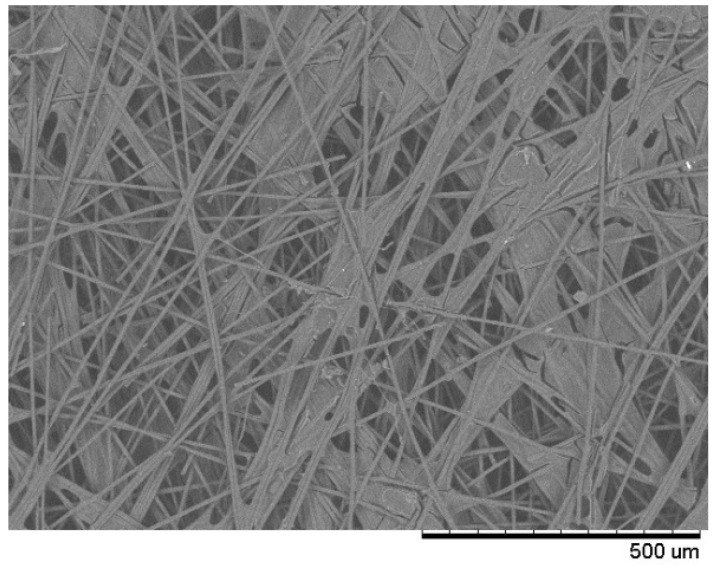
SEM (Hitachi^®^ TEM3000, Tokyo, Japan, 15 kV) image of AVCARB GDL fibbers.

**Figure 2 membranes-12-00212-f002:**
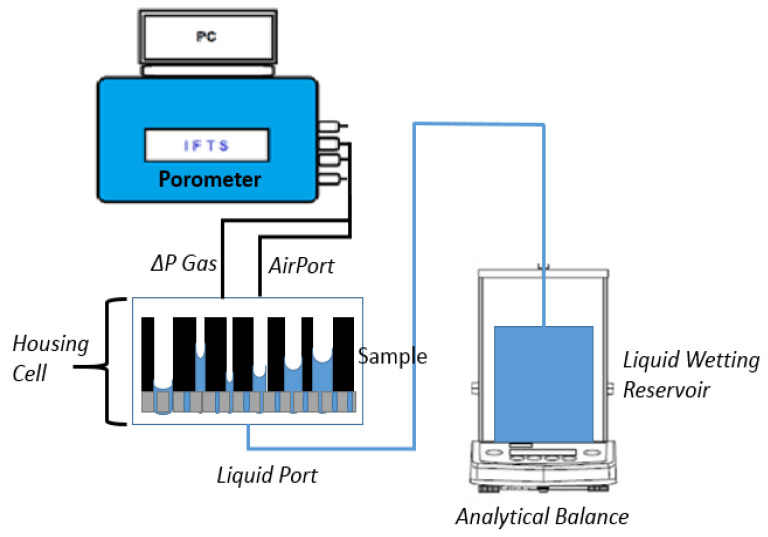
Schematic diagram of Liquid Extrusion Porometer. System setup.

**Figure 3 membranes-12-00212-f003:**
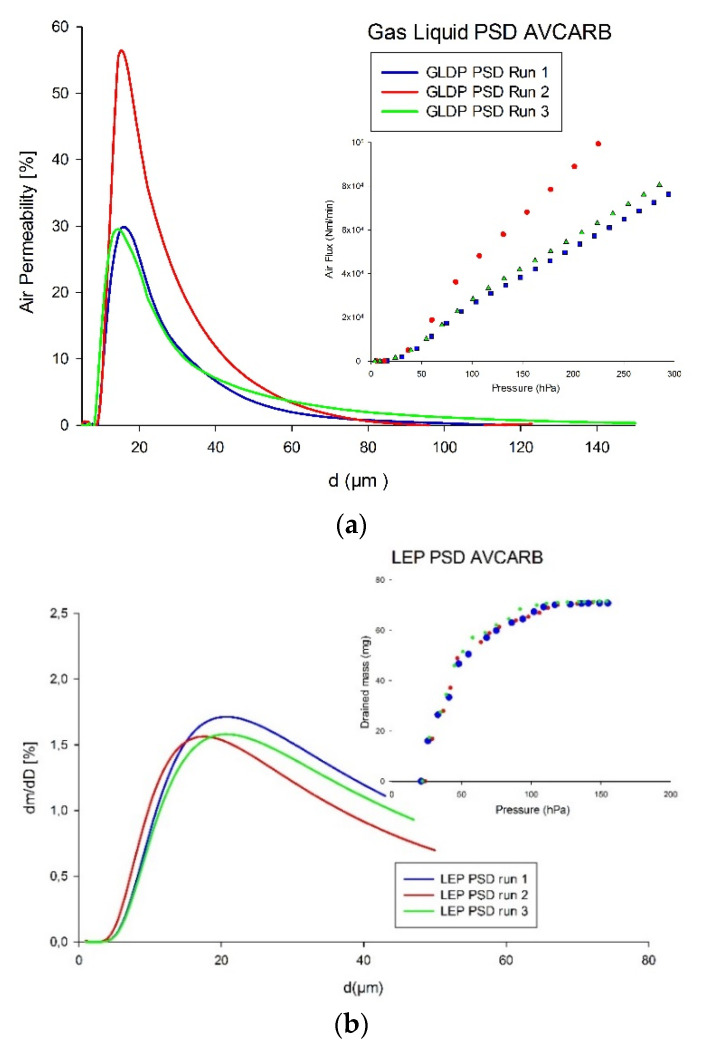
Experimental curves and resulting PSDs of AVCARB GDL by (**a**) GLDP; (**b**) LEP.

**Figure 4 membranes-12-00212-f004:**
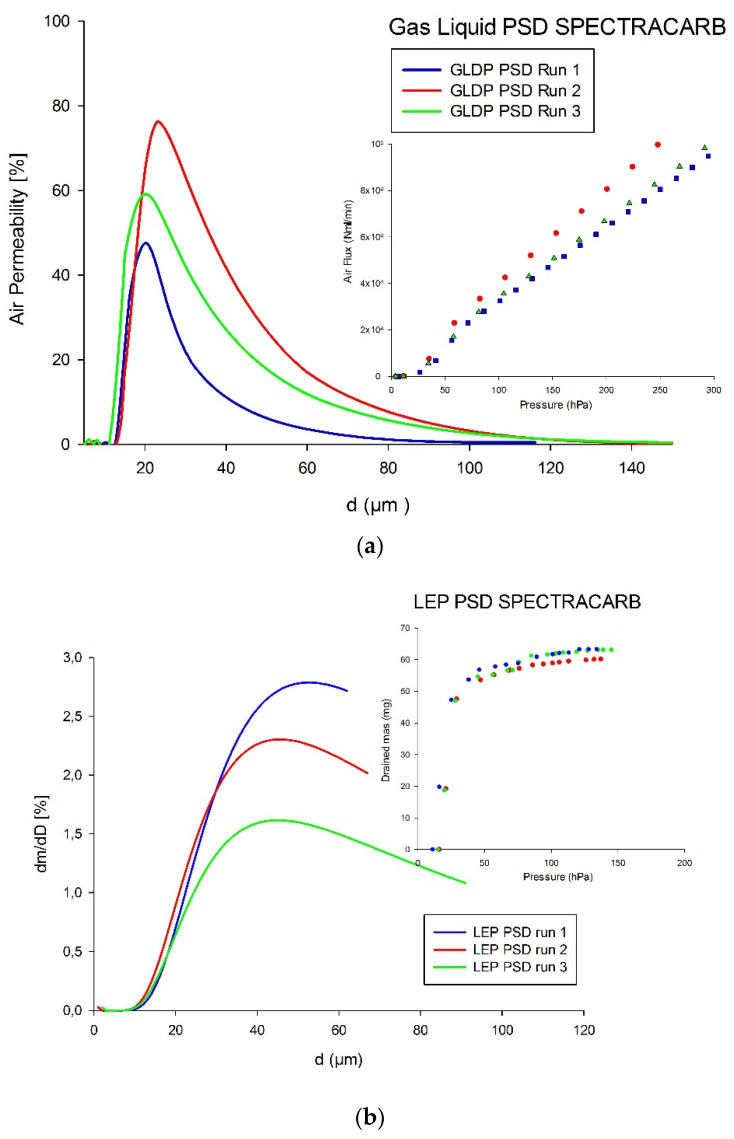
Experimental curves and resulting PSDs of SPECTRACARB GDL by (**a**) GLDP; (**b**) LEP.

**Figure 5 membranes-12-00212-f005:**
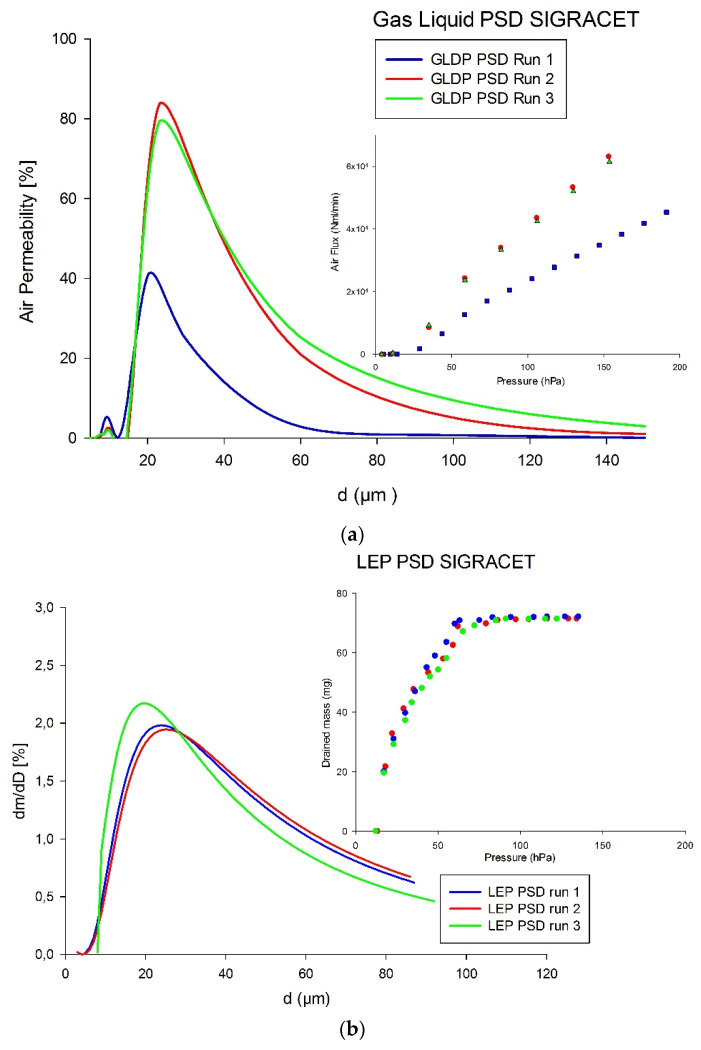
Experimental curves and resulting PSDs of SIGRACET GDL by (**a**) GLDP; (**b**) LEP.

**Table 1 membranes-12-00212-t001:** Names and models of the three GDL media analyzed.

Manufacturer/Model
AVCARB MGL280, Lowell, MA, USA
SIGRACET 39AA, Seattle, WA, USA
SPECTRACARB 2050A-0850, Shelton, CT, USA

**Table 2 membranes-12-00212-t002:** Mean pore diameters (*d_avg_*) and most frequent diameter (*d_mod_*) obtained via GLDP and LEP for the three GDL samples tested.

GDL	*d_avg_* (µm) GLDP	*d_avg_* (µm) LEP	*d_mod_* (µm) GLDP	*d_mod_* (µm) LEP
AVCARB	19.4 ± 1.5%	26.3 ± 3.0%	14.2 ± 5.1%	19.6 ± 9.2%
SPECTRACARB	26.6 ± 6.9%	47.3 ± 12.9%	22.6 ± 4.3%	41.0 ± 5.1%
SIGRACET	30.4 ± 29%	42.1 ± 0.9%	22.2 ± 6.0%	22.8 ± 12.8%

## Data Availability

Not applicable.

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
