# Peer review of "Characterization of Commercial Gas Diffusion Layers (GDL) by Liquid Extrusion Porometry (LEP) and Gas Liquid Displacement Porometry (GLDP)"

_membranes, 2022, doi:10.3390/membranes12020212_

Round 1
Reviewer 1 Report
This research uses Gas Liquid Displacement Porometry (GLDP) and Liquid Extrusion Porometry (LEP) techniques analyzed three commercial GDL, the pore size distributions (PSD) and mean pore diameters were obtained through experiments. The results of two different techniques were compared together and several conclusions were drawn. The whole article is good, but some places still need to be improved:
- There are some spelling or grammatical mistakes in the article, for example: (1) Abstract: “GDL and LEP”, may be “GLDP and LEP”. (2) Introduction section: “iintrusion”. (3) Section 2.3 “supply a stable has pressure”.
- The format of introduction part is not right. It is divided into different paragraphs and lack logicality.
- The table 1 has no information. It is all the same except the manufacturer.
- Why the permeability did not have the Units in some figures.
- The results of this paper focus on the permeability and mean pore diameters, which are mainly the properties of the GDL material itself, rather than the performance impact of GDL in fuel cells.
Author Response
Reviewer 1
This research uses Gas Liquid Displacement Porometry (GLDP) and Liquid Extrusion Porometry (LEP) techniques analyzed three commercial GDL, the pore size distributions (PSD) and mean pore diameters were obtained through experiments. The results of two different techniques were compared together and several conclusions were drawn. The whole article is good, but some places still need to be improved:
Author: Thanks so much for your comments and suggestions, according to them we have done our best to improve the manuscript.
- There are some spelling or grammatical mistakes in the article, for example: (1) Abstract: “GDL and LEP”, may be “GLDP and LEP”. (2) Introduction section: “iintrusion”. (3) Section 2.3 “supply a stable has pressure”. Done
- The format of the introduction part is not right. It is divided into different paragraphs and lack logicality.
Author: We have tried to go sequentially in this introduction, from general FC to GDL, remarking its role in the FC and its expected characteristics. Finally we have focussed in how to determine such characteristic parameters focussing on what we consider our goal, comparison of two useful methods to determine the porous characteristics of GDL. In the revised version we have tried to make more fluid the transition of these steps to the final explanation of the aim of our work.
3. The table 1 has no information. It is all the same except the manufacturer.
Author: Certainly, table 1 had no useful information. We have reduced to the names of the different GDL models.
4. Why the permeability did not have the Units in some figures.
Author: Done. In both techniques (GLDP and LEP), we have represented the PSD as contribution to the total air permeability or mass differential increment of each data point. Accordingly the units in both plots are %, then dimensionless. Moreover we have improved the clarity of the figures to have all of them in same format and make easier the comparison.
5. The results of this paper focus on the permeability and mean pore diameters, which are mainly the properties of the GDL material itself, rather than the performance impact of GDL in fuel cells.
Author: You are right, but we focussed (as stated in the aim of the work) on the comparison of two porosity characterization membranes (both able to be used with GDLs) and trying to see which one is more accurate and/or precise in such PSD characterization. This information should be followed, of course, by determination of other more performance-related parameters (structural, functional, or even constitutional) but this cannot be supplied by the techniques here analyzed.
Reviewer 2 Report
The authors use two techniques, Gas Liquid Displacement Porometry (GLDP) and Liquid Extrusion Porometry (LEP) to characterize the pore structure of three types of carbon fiber paper-based gas diffusion layers (GDLs) for fuel cells.
It is not clear what the authors try to achieve. The conclusions are either trivial or nonsense:
“Obviously, the experimental behaviour of each technique and, consequently, the resulting PSD could be slightly different depending on …”
“The results are very interesting, showing a nice agreement between…”
“In that sense, it should be remarked that always (for the three filters analyzed and both mean or mode values), the resulting curves from LEP are slightly shifted to higher pore sizes than those coming from GLDP”
Other recommended editorial changes:
- Rephrase the following: ” Three carbon based GDL presenting a highly rigid microstructure of interconnected pores of several manufactured which being hardly considered a bundle of parallel pores were analyzed.” It is not clear what the authors tried to say.
- On page 2, the authors state that an MEA consists of “a gas diffusion layer (GDL), a catalyst layer, and a proton exchange membrane”. In reality, an MEA consists of two GDLs, 2 catalyst layers and one proton exchange membrane.
- “design” instead of “deign”.
- “bipolar plate” instead of “bi-component plate”.
- On page 2, the authors state that “a microporous layer (MPL) […] is made from a fibrous carbon-based paper material”. This is not correct. An MPL is made of a mixture of carbon or graphite powder and poly(tetra)fluoroethylene, or other polymer used as binder and hydrophobic agent. This mixture is wetted with a solvent and turned into a “mud” that is impregnated into the GDL pores on the side that faces the catalyst layer. After impregnation the solvent is evaporated, and the “mud” solidifies.
- “intrusion” instead of “iintrusion”.
- The authors use a few times, including in the title of section 2.1 the syntagm “GDL filters”. GDLs do not have the role of filtrating anything. Use instead “GDL”.
- Referring to Table 1, the authors state that “The information available on the membranes and characterization method are given in Table 1”. There is no information on any membranes, and no information on any characterization method in Table 1. Rephrase.
- On page 5 the authors refer to 2, but the first equation in the manuscript is numbered “Eq.3”.
Author Response
Reviewer 2
The authors use two techniques, Gas Liquid Displacement Porometry (GLDP) and Liquid Extrusion Porometry (LEP) to characterize the pore structure of three types of carbon fiber paper-based gas diffusion layers (GDLs) for fuel cells.
It is not clear what the authors try to achieve. The conclusions are either trivial or nonsense:
“Obviously, the experimental behaviour of each technique and, consequently, the resulting PSD could be slightly different depending on …”
“The results are very interesting, showing a nice agreement between…”
“In that sense, it should be remarked that always (for the three filters analyzed and both mean or mode values), the resulting curves from LEP are slightly shifted to higher pore sizes than those coming from GLDP”
Author: Thanks so much for your thoughtful comments and suggestions. We have done our best to improve the manuscript, based on your comment and we present a new version that we hope could be considered clearer and more interesting for scientific community. In particular, conclusions have been revised focussing on the comparison of both techniques for the characterization of GDLs.
Other recommended editorial changes:
- Rephrase the following: ” Three carbon based GDL presenting a highly rigid microstructure of interconnected pores of several manufactured which being hardly considered a bundle of parallel pores were analyzed.” It is not clear what the authors tried to say.
Author: We have changed the phrase to render it hopefully clearer.
- On page 2, the authors state that an MEA consists of “a gas diffusion layer (GDL), a catalyst layer, and a proton exchange membrane”. In reality, an MEA consists of two GDLs, 2 catalyst layers, and one proton exchange membrane.
Author: Thanks to point out the mistake, we have changed it accordingly.
- “design” instead of “deign”. Done
- “bipolarplate” instead of “bi-component plate”. Done
- On page 2, the authors state that “a microporous layer (MPL) […] is made from a fibrous carbon-based paper material”. This is not correct. An MPL is made of a mixture of carbon or graphite powder and poly(tetra)fluoroethylene, or other polymer used as binder and hydrophobic agent. This mixture is wetted with a solvent and turned into a “mud” that is impregnated into the GDL pores on the side that faces the catalyst layer. After impregnation the solvent is evaporated, and the “mud” solidifies.
Author: Thanks, we referred mostly to the composition of MPS, but we have changed the phrase according to your suggestion. In fact the GDL we analysed are not composed of MPS and MPL while they present only a porous layer made from carbon paper with no polymer addition.
- “intrusion” instead of “iintrusion”. Done
- The authors use a few times, including in the title of section 2.1 the syntagm “GDL filters”. GDLs do not have the role of filtrating anything. Use instead “GDL”. Done
- Referring to Table 1, the authors state that “The information available on the membranes and characterization method are given in Table 1”. There is no information on any membranes, and no information on any characterization method in Table 1. Rephrase.
Author: Done, table has been reduced, accordingly, to include only the names of the GDL samples and manufacturers
- On page 5 the authors refer to 2, but the first equation in the manuscript is numbered “Eq.3”.
Author: Done, also changed the incorrect numbering of Eq. 4
Round 2
Reviewer 1 Report
The authors have revised depend on my suggestion. I think that it can be published.
Author Response
Thanks so much,
We want to acknowledge both referees for their comments and suggestions that resulted in an improved manuscript.
Reviewer 2 Report
I can recommend the manuscript for publication after some minor revision:
Page 2: instead of “Methanol Fuel Cell (DMFC)”, use “Direct Methanol Fuel Cell (DMFC)”.
Page 2: As I mentioned in my original review, MPS (micro-porous layers) are not made of a fibrous carbon-based paper mixed with a polymer binder, but from carbon black (powder) mixed with a polymer binder. This carbon black (for example Vulcan, Katjen, etc.) has particle sizes in the order of 20-40 nm.
Author Response
Thank you so much
We want to acknowledge both referees for their comments and suggestions that resulted in an improved manuscript.
Reviewer 2
- Page 2: instead of “Methanol Fuel Cell (DMFC)”, use “Direct Methanol Fuel Cell (DMFC)”.
Author: Done
- Page 2: As I mentioned in my original review, MPS (micro-porous layers) are not made of a fibrous carbon-based paper mixed with a polymer binder, but from carbon black (powder) mixed with a polymer binder. This carbon black (for example Vulcan, Katjen, etc.) has particle sizes in the order of 20-40 nm.
Author: Done, We have changed the phrase to include referees' comment.